

# Systematic identification of smORFs in domestic silkworm (*Bombyx mori*)

Linrong Wan[1,2,*], Wenfu Xiao[1,*], Ziyan Huang[3,4], Anlian Zhou[1], Yaming Jiang[1], Bangxing Zou[1], Binbin Liu[1], Cao Deng[3,4] and Youhong Zhang[1]

[1] Sericultural Research Institute,Sichuan Academy of Agricultural Sciences, Nanchong, Sichuan, China
[2] College of Agronomy, Sichuan Agricultural University, Chengdu, Sichuan, China
[3] Research and Development Center, LyuKang, Chengdu, Sichuan, China
[4] Departments of Bioinformatics, DNA Stories Bioinformatics Center, Chengdu, Sichuan, China
[*] These authors contributed equally to this work.

## ABSTRACT

The silkworm (*Bombyx mori*) is not only an excellent model species, but also an important agricultural economic insect. Taking it as the research object, its advantages of low maintenance cost and no biohazard risks are considered. Small open reading frames (smORFs) are an important class of genomic elements that can produce bioactive peptides. However, the smORFs in silkworm had been poorly identified and studied. To further study the smORFs in silkworm, systematic genome-wide identification is essential. Here, we identified and analyzed smORFs in the silkworm using comprehensive methods. Our results showed that at least 738 highly reliable smORFs were found in *B. mori* and that 34,401 possible smORFs were partially supported. We also identified some differentially expressed and tissue-specific-expressed smORFs, which may be closely related to the characteristics and functions of the tissues. This article provides a basis for subsequent research on smORFs in silkworm, and also hopes to provide a reference point for future research methods for smORFs in other species.

# INTRODUCTION

Small open reading frames (smORFs), like genes, are important categories of genomic elements that subvert our understanding of genome coding potential (*Basrai, Hieter & Boeke, 1997*). At first, smORFs were considered nontranscriptable and untranslatable due to their length, which is less than 100 codons (*Wu et al., 2019*). However, recent studies have found that millions of smORF sequences exist in the eukaryotic genome and can be transcribed into RNA. They can be divided into "coding" function (producing bioactive peptides) and "non-coding" regulatory function (participating in translation mechanism) RNAs (*Couso & Patraquim, 2017*). Therefore, a small portion of smORFs have the potential to be translated into polypeptides (*Wu et al., 2019*; *Ladoukakis et al., 2011*). These peptides with lengths less than 100 amino acids are called micro-peptides or smORF-encoded peptides (SEPs) (*Chen et al., 2021*). Further studies on micro-peptides based on bioinformatics and high-throughput sequencing technology have found that micro-peptides not only are highly conserved throughout evolution (*Ladoukakis et al.,*

Corresponding authors
Cao Deng, dengcao@dnastories.com
Youhong Zhang, sczhangyh@126.com

*2011*) but also play an important regulatory role in a series of processes such as biological development (*Sanchez-Ortiz, 2017*; *Read et al., 2019*), metabolism (*Makarewich et al., 2018*; *Stein et al., 2018*), and cancer incidence (*Pang et al., 2020*; *Wu et al., 2020*; *Li et al., 2020*). Due to the important biological function of small open reading frames, it has gradually become a popular topic of research.

Domestic silkworms (*Bombyx mori*) have been raised for more than 5,000 years for silk production (*Wan et al., 2021*; *Fang et al., 2015*) and are now used for commercial production of important biomedical and industrial bio-materials based on genetic engineering (*Ude et al., 2014*; *Cao & Zhang, 2016*), in addition to being used as food in some Asian countries (*He et al., 2021*). Silkworms are also similar to humans in terms of their sensitivities to pathogens and the comparable effects of drugs on them, and their advantages for research are their low cost of maintenance, few ethical constraints, and no biohazard risks (*Nwibo et al., 2015*; *Nouara, Lü XMLAMP Chen, 2018*). Hence, the silkworm has long been recognized as an excellent model organism, similar to *Drosophila*, for studying physiology, biochemistry, developmental biology, neurobiology, and pathology (*Kawamoto et al., 2019*; *Tong et al., 2022*). The study of small open reading frames in silkworm may play an important role in promoting the development of the sericulture industry.

To date, small open reading frames have been thoroughly studied mainly in *Drosophila*. For example, *Ladoukakis et al. (2011)* systematically screened and identified small open reading frames in the *Drosophila* genome, the functions of which for some smORFs were further verified by *Magny et al. (2013)* and *Pueyo et al. (2016)*. However, our current understanding of smORFs in silkworm is insufficient. The cloning and functional verification of a few smORFs in *B. mori* are almost entirely based on these smORFs that have been well-studied in *Drosophila* (*Cao et al., 2018*; *Zhu et al., 2019*). More importantly, we still lack some systematic and comprehensive studies on the identification and analysis of smORFs in *B. mori*. In most gene annotation procedures, some basic principles are generally followed to ensure the accuracy of the results in identifying open reading frames (ORFs). For example, the minimum length cut-off point ($\geq$100 aa) is usually used to prevent false annotation of ORFs (*Jackson et al., 2018*; *Hanada et al., 2007*), and the smaller ORF nested in the larger ORF is usually not annotated as a gene alone (*Wu et al., 2019*). However, smORFs are different from ORFs in sequence length and their theoretical length can be limited from 2 to 100 codons, which makes it difficult to annotate them by traditional gene annotation methods (*Wu et al., 2019*; *Hanada et al., 2007*).

Based on transcriptomic sequencing datasets, the present work systematically and comprehensively identified and analyzed smORFs in the silkworm genome by integrating previous influential and reliable methods, so as to provide a basis for subsequent research on small open reading frames in silkworm. We further hope to provide reference point for future development of research methods for smORFs in other species.

## MATERIALS & METHODS

### Genomic and transcriptomic data

The silkworm genome and annotation data were downloaded from SilkBase (http://Silkbase.ab.a.u-tokyo.ac.jp/cgi-bin/download.cgi), which was published in 2019 and based
on 140 × deep sequencing of long (PacBio,Menlo Park,USA) and short (Illumina,San Diego,USA) readings. The new genome annotated more RNA-seq and protein data, resulting in higher quality genome assembly and more accurate gene model than the previous version (*Kawamoto et al., 2019*). The genomes and annotations of *Heliconius melpomene* (GCA_000313835.2), *Melitaea cinxia* (GCA_905220565.1), *Operophtera brumata* (GCA_001266575.1), and *Drosophila melanogaster* (GCF_000001215.4) used in this article are all downloaded from National Center for Biotechnology Information (NCBI).

Transcriptomic raw data were downloaded from previous studies and each sample has three biological replicates (see Table S1 for details). The RNA-seq data of *B. mori* strain p50T were downloaded from the NCBI Bioproject PRJDB8614, which contained 10 tissues/subparts from 3rd day of 5th instar larvae measured by Illumina NovaSeq6000 (*Yokoi et al., 2021*). The RNA-seq data of *B. mori* strain o751 (wild-type) were taken from Bioproject PRJDB4976, including five tissues from 3rd-day 5th instar larvae measured by an Illumina HiSeq 2000 (*Ichino et al., 2018*; *Kikuchi et al., 2017*; *Kobayashi et al., 2019*).

## Sequencing alignment and transcript assembly

The original sequence obtained by sequencing contains low-quality reads and adaptor sequences; however, the subsequent analysis must be based on clean reads. To obtain high-quality clean reads, the raw sequencing reads were filtered using Trimmomatic software (version 0.39) (*Bolger, Marc & Bjoern, 2014*) with the following steps: First, reads with adaptor sequences were removed. Then, reads containing more than 30% of low-quality bases (Q < 20) or containing more than 3% of ambiguous "N" were discarded. The reads were also trimmed where the four-bases-window had an average quality lower than 20.

Since the genome and annotation information of this species is available, it is better to use genome and annotation information to assist transcript assembly. After the filtering steps, the clean reads from each sample were aligned to the updated genome assemblies (http://silkbase.ab.a.u-tokyo.ac.jp/cgi-bin/download.cgi) using HiSAT2 (version 2.0.4) (*Kim, Langmead & Salzberg, 2015*) with default parameters, which generated BAM files for downstream analysis. The statistical power of this experimental design, calculated in RNASeqPower (https://rodrigo-arcoverde.shinyapps.io/rnaseq_power_calc/) is 0.983.

The mapped data, the BAM files, were each assembled to transcriptome data by StringTie (version 2.1.7) (*Pertea et al., 2015*), and the minimum read coverage, minimum input transcript length, and minimum locus gap separation were set to 10X, 30 bp, and 1 bp, respectively. Then, StringTie-merge was used to filter the 45-transcriptome data under the following conditions: input transcript coverage ≥ 10, input transcript FPKM ≥ 1, isoform fraction ≥ 0.1, locus gap separation ≥ 1. Finally, the data was merged into a new reference transcriptome.

## Identification of smORFs

The identification process for smORFs is shown in Fig. 1A. To identify smORFs in *B. mori*, it is essential to obtain the silkworm-specific Kozak sequences. First, the existing protein sequences of *B. mori* were aligned to the SwissProt (*Duvaud et al., 2021*) database

(https://www.expasy.org/resources/uniprotkb-swiss-prot), and the alignment results were retained according to the conditions of identity ≥ 30% and coverage ≥ 30% to identify the proteins with high reliability. Second, a set of Kozak sequences with a total length of 14 bp was obtained by extracting the upstream 9 bp (−9 to −1) and downstream 3 bp (+1 to +3) sequences of translation initiation sites. Finally, according to the rules of the Kozak sequence (*Kozak, 2002*), the first codon does not need to be ATG, it may have a variety of forms, namely the first base substitution: TTG, GTG, CTG; the second base substitution: AAG, ACG, AGG; the third base substitution: ATT, ATC, ATA (As shown in Fig. 1A).

Transcript-based *de novo* annotation: Using software ORFfinder (version: 0.4.1) downloaded from NCBI (https://www.ncbi.nlm.nih.gov/orffinder/) to predict the smORFs in each transcription sequence, the parameter was set to "-ml 5 -s 2 -strand plus," that is, the minimum length is 5, located in the sense chain, and the starting codon is a meaningful codon. The annotated results retain results with less than 300 bp (100 amino acids) in length and must have a stop codon. In addition, according to the KOZAK rule, it is not appropriate to set only to normal ATG, so the final merged rule are starting codon is an ATG or KOZAK variant, less than 100 amino acids and with a termination codon.

The *de novo* annotation of conservative regions was based on genome-wide alignment: First, the silkworm genome was aligned with *Heliconius melpomene*, *Melitaea cinxia*, *Operophtera brumata,* and *Drosophila melanogaster* genome using LastZ software (*Harris, 2007*) (version: 0.4.1, default parameters), and then the conservative regions in at least two species above silkworm were calculated and extracted with an in-house script using the software ORFfinder (version: 0.4.1) downloaded from NCBI (https://www.ncbi.nlm.nih.gov/orffinder/) to predict the smORFs in each conservative region. The Except-Strand parameter was set to "both," the other parameters and filtering conditions were consistent with the above.

## Classification of smORFs

We conducted research on classification by confidence level of smORFs from two aspects. One was the prediction of its coding potential and the other was homologous alignment in the relevant database, CPPred-sORF script (*Tong et al., 2020*) (http://www.rnabinding.com/CPPred-sORF/), which was used to predict the potential of *de novo* annotated smORF sequences. According to the length of protein sequences translated by smORFs, it was divided into a long sequence set (>15 aa) and a short sequence set (≤ 15 aa). Sequences in the long sequence set were aligned to the reference smORF database, which has integrated the current reliable smORF databases: sORFs. Org (*Volodimir, Van Criekinge & Gerben, 2018*) (http://sorfs.org/), OpenProt (*Brunet et al., 2019*) (https://openprot.org), SmProt (*Hao et al., 2018*) (http://bigdata.ibp.ac.cn/SmProt/) and for proteins less than 100 amino acids in Refseq (*O'Leary et al., 2016*) (https://www.ncbi.nlm.nih.gov/refseq/), and Swissport (*Duvaud et al., 2021*) databases. Those sequences that were collected in Reference smORF database with *E* value ≤ 0.15 were retained, according to the study of *Ladoukakis et al. (2011)*. For the sequences in the short sequence set, we directly confirmed whether they have been collected in smORFs using Org (*Volodimir, & Gerben, 2018*) database.

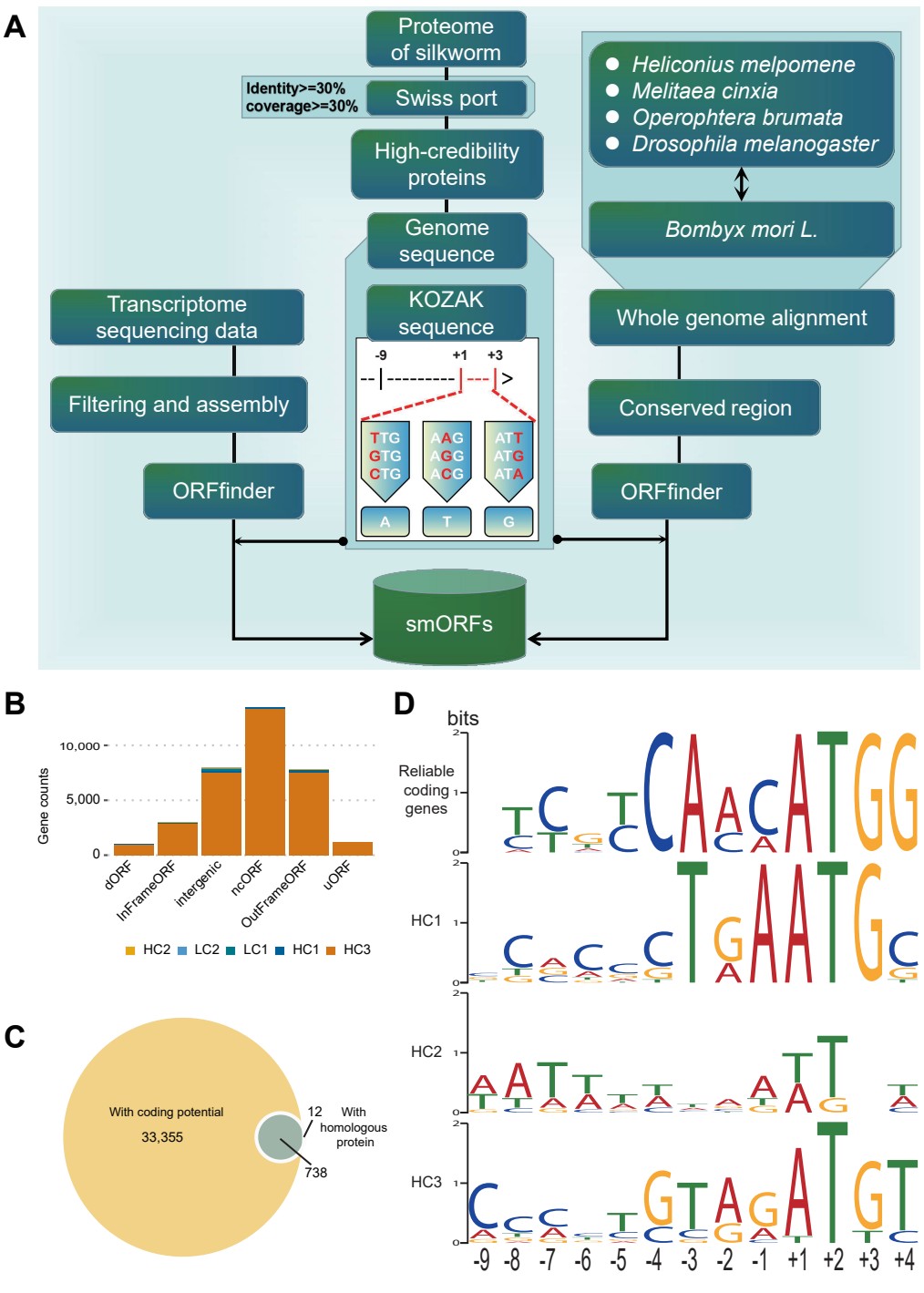

**Figure 1** **Identification and classification of smORF.** (A) Identification pipeline of smORF. (B) Number of smORFs per confidence level classifications. (C) The composition of supporting evidence types of high confidence smORFs. (D) Kozak motif logo of reliable coding genes and high confidence smORFs.

Then, we classified smORFs according to their confidence level. These smORFs were divided into high confidence (HC1–HC4) and low confidence (LC1-2). The classification criteria are shown in Table S2 , where categories are ranked by level of credibility.

We further classified smORFs according to the relationship between smORFs and genes, then ranked the priority in each category. smORFs were divided into InFrameORF, OutFrameORF, uORF, dORF, ncORF (including lncORF, miORF, and circORF), intronORF, and intergenicORF. Definitions and diagrams are shown in Table S3 and sorted by their priority.

## Quantification of smORF expression levels in different tissues

To analyze the differential and tissue-specific expression of smORFs, quantitative expression is a prerequisite. HTSeq (*Anders, Pyl & Huber, 2015*) (version 0.11.3) is one of the most commonly used quantitative software packages, which requires BAM files and annotation files as input data. BAM files were generated from the "Sequencing alignment and transcript assembly" section by HISAT2, and annotation files were generated from a combination of the downloaded genome annotation files and our smORF annotation files. The parameters of HTSeq software were " $-$ f bam $-$ s no $-$ r pos $-$ a 10 $-$ m union $-$ t exon." The count expression matrix generated by HTSeq-count was used to estimate smORF expression levels. Specifically, if the coordinate of smORF overlaps with the parent gene, the expression of this gene was regarded as the expression of smORF. On the contrary, if the coordinate of smORF did not overlap with any genes, the coordinate region of the smORF was quantified independently. The correlation was calculated by script PtR in Trinity RNA-Seq (*Haas et al., 2013*; *Grabherr et al., 2011*) (version 2.11.0) with the same count expression matrix.

## Differentially expressed and tissue-specific expression analysis

Differential expression analysis was performed using the script run-DE-analysis.pl from Trinity RNA-Seq (version 2.11.0) with input data from expected counts generated by HTSeq-count software. The Analyze-diff-expr.pl script in Trinity RNA-Seq was used for subsequent expression analysis. The threshold for significantly-differential expression was set to FDR $\leq$ 0.05 and log2(fold change) $\geq$ 2.

The tspex (*Antonio et al., 2021*) (version: 0.6.2) is a tissue-specificity calculator software for calculating a variety of tissue-specificity metrics from gene expression data. The tissue-specificity index (TSI), which was calculated by tspex, was used for the assessment of tissue-specific expression of genes and smORFs in various samples following the study of *Julien et al. (2012)*. Preprocessing methods of the input matrix are essential (*Kryuchkova-Mostacci & Robinson-Rechavi, 2017*). The following steps were specifically employed: 1. the value of RPKM less than 1 was set to 0; 2. then log10(FPKM) was processed to remove the smORF with 0 expression level; 3. the mean value of biological replicate samples in the same tissue was calculated and used as the expression input matrix. The range of TSI values was between 0 and 1 and was positively correlated with the gene expression specificity in the tissue. The threshold was set to 0.8 according to *Kryuchkova-Mostacci & Robinson-Rechavi (2017)*. For flexibility, we also used 0.85, 0.9, 0.95, and 1 as thresholds to calculate smORFs specifically expressed in each tissue.
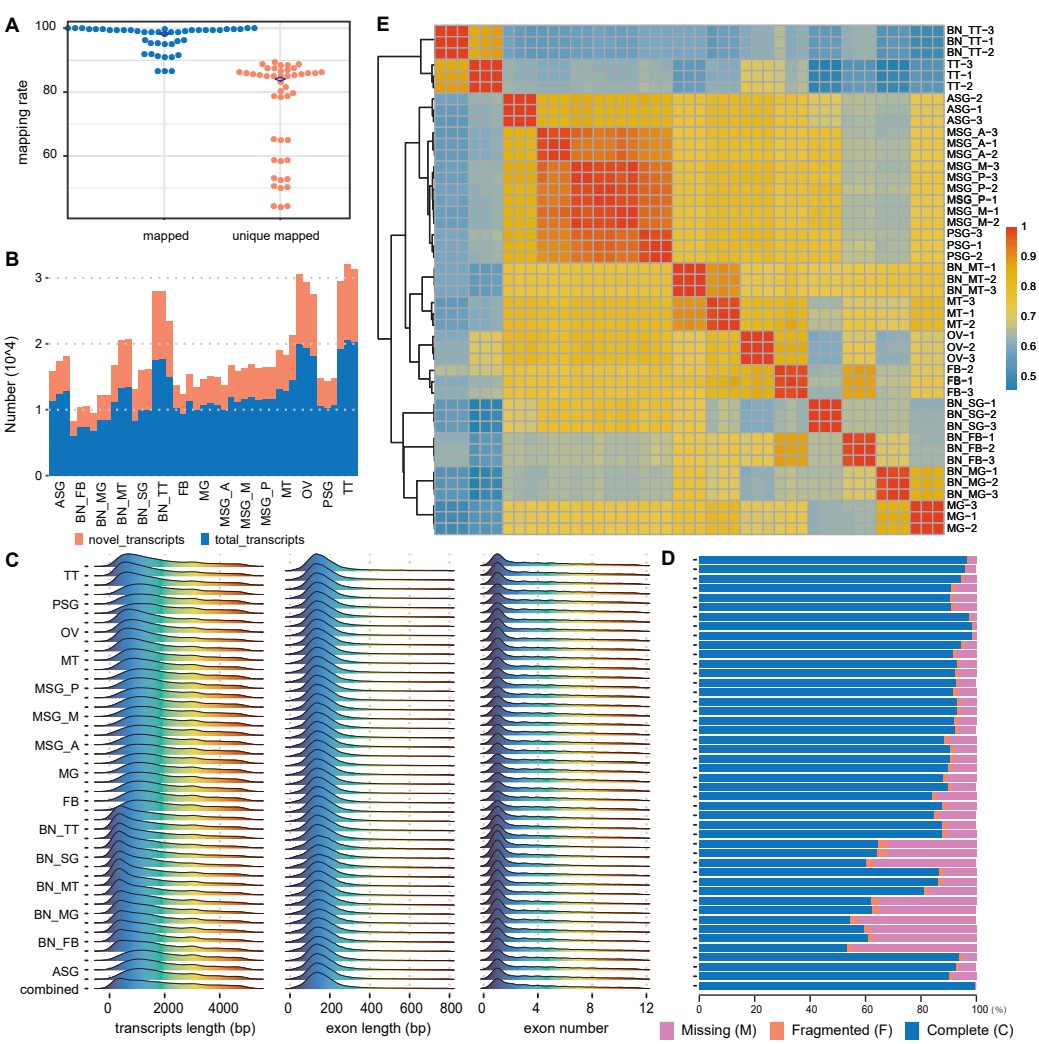

**Figure 2** **Summary of alignments and transcript assemblies using transcriptomic sequencing.** (A) Mapping rate and unique mapped rate of transcriptome samples. (B) Number of gene loci (known, novel) in each sample. (C) Distribution of transcript length, exon length, and exon number. (D) Integrity evaluation of sample annotation and merged annotation with BUSCO. (E) Heatmap of sample correlation matrix based on gene expression.

# RESULTS

## Data prepossessing, reads alignment, and transcript assembly

Transcriptomic sequencing of a range of silkworm tissue sources revealed many novel gene sequences (Fig. 2). The transcriptome data downloaded in this paper include 10 different major tissues of silkworms and silk gland tissues can be divided into anterior, middle, posterior, and whole silk gland tissues (Table S1). After data filtering, these samples retain only high-quality parts of the original data, with an average retention rate of about 85%. In these samples, the lowest proportion is 56.75%, and the highest proportion is 97.89% (Table S4).

According to the alignment results of clean reads through HISAT2, in each sample the highest proportion of reads that could be aligned to the reference genome was 98.95% with the lowest at 85.65%, averaging 94.49%. The highest proportion of reads that could only be aligned to the reference genome sequence once was 88.45%, the lowest of which was 43.27%, with an average of about 67.25% (Fig. 2A, Table S5).

According to the assembly and merge results of clean reads through StringTie, novel genes and novel transcripts were found in all samples. The ratio of novel genes ranged from 24.09% to 59.18%, with an average of 37.68%; the ratio of novel transcripts was between 31.21% and 62.20%, with an average of 45.54% (Fig. 2B, Table S6).

The transcript length was mainly distributed between 200 and 300 bp, single exon length was distributed between 100 and 200 bp, and the number of transcript exons was concentrated between 1 and 2 (Fig. 2C). In the subsequent statistical analysis, we classified and counted the variable number of isoforms of genes, and the results showed that the number of isoforms was negatively correlated with the number of genes category (Table S7 ). According to the results of the BUSCO annotation integrity verification, for each sample based on 1,013 conserved genes, the integrity evaluation of sample annotation and merged annotation results were not completely in agreement. However, after the merger, the integrity reached 1,010 complete genes and the fragment size was three genes, without missing genes (Fig. 2D).

The correlation analysis was performed according to the gene expression of each sample. The results are shown in Fig. 2E. The correlation between different replicates of the same sample was higher than that between samples. Concurrently, the correlation between samples with the same or similar sources was also high; for example, the correlation between MSG_A (anterior part of the middle silk gland), MSG_M (middle part of the middle silk gland), MSG_P (posterior part of the middle silk gland), and PSG (posterior silk gland) was high.

## Identification of smORFs

To identify potential smORFs, we established the analysis pipeline shown in Fig. 1A. By *de novo* annotation of RNA-seq, conserved region prediction, prediction of coding and screening, and construction of silkworm-specific Kozak sequences, 34401 smORFs were identified in *B. mori*.

## Classification of smORFs

According to the supporting evidence of each smORFs, 34401 smORFs can be classified into high confidence and low confidence levels as shown in Table 1. Among the 34105 smORFs in the high confidence level (HC), 33355 smORFs were identified as smORFs with coding potential by CPPred-sORF software and 750 smORFs were in or with homologous sequences in the joint database. Of these, 738 smORFs were supported by both lines of evidence (Fig. 1C). A total of 296 smORFs were identified in low-confidence level (LC), of which 215 were supported by *de novo* transcript prediction only and 81 by evidence of evolutionary conservation only.

Subsequently, the Kozak sequences of reliable coding genes (RGs) and smORFs of high confidence levels (HC1-3), *i.e.,* 9 bp upstream to 3 bp downstream of the starting

**Table 1  Number of smORFs in each confidence level and their classifications.**

|       | InFrameORF | OutFrameORF | dORF | Intergenic ORF | ncORF  | uORF  | Total  |
|-------|------------|-------------|------|----------------|--------|-------|--------|
| HC1   | 165        | 259         | 10   | 130            | 161    | 13    | 738    |
| HC2   | 1          | 1           | –    | 10             | –      | –     | 12     |
| HC3   | 2,855      | 7,534       | 982  | 7,529          | 13,286 | 1,169 | 33,355 |
| LC1   | 1          | –           | –    | 214            | –      | –     | 215    |
| LC2   | 4          | 13          | –    | 59             | 1      | 4     | 81     |
| total | 3,026      | 7,807       | 992  | 7,942          | 13,448 | 3,026 | 34,401 |

transcription site, were analyzed as shown in Fig. 1D. According to the results of statistical analysis, RGs, HC1 and HC3 are conservative at specific positions. Among them, RGs has only one possible base composition at six sites ($-4, -3, +1, +2, +3, +4$), while HC1 and HC3 have five and one sites respectively. The statistical analysis results of the initiation codons show that both RGs and HC1 are conventional ATG, while the initiation codons of HC2 and HC3 have different types besides ATG, such as ATT, TTG, AGG, etc. In general, the Kozak sequence polymorphism of RGs and HCs was mainly concentrated between -9bp and -5bp.

These identified 34,401 smORFs can also be classified according to their different positions in the genome and different relative positions within their parent genes (Table S3). In this study, smORFs are divided into six categories: InFrameORF, OutFrameORF, dORF, intergenicORF, ncORF, and uORF. Among these categories, there were 13,448 non-coding ORFs, accounting for 39.09% of the total number, followed by 7,942 Intergenic ORFs and 7,807 OutFrameORFs, accounting for 23.09% and 22.69% of the total, respectively. The lowest number was found in dORFs (992), which were located downstream of the parent gene, only accounting for 2.88%. However, the proportion of different types of smORFs was not completely related to their overall expression. For example, the proportion of dORFs in the identified smORFs was the smallest, but the overall expression was the highest. The proportion of Intergenic ORFs in the identified smORFs was only second to that of non-coding ORFs, but their overall expression level was the lowest.

## Differential expression of smORFs

The correlation analysis of the samples was performed according to the identified smORF expression. As shown in Fig. 3A, the correlation analysis was consistent with the gene correlation analysis: the correlation between different repetitions of the same sample was high and the correlation between samples with the same or similar sources was also high.

To explore the expression of smORFs in different tissues of silkworm, the expression of smORFs in different samples was quantitatively analyzed using Trinity RNA-Seq software. The average of three biological repeat expression levels of smORFs in each tissue was taken as the expression level of smORFs in the tissue. With TPM value as a reference, the expression of all genes and all identified smORFs was analyzed. It was found that the overall gene expression was higher than that of smORFs, and the $P$-value was less than $2.22e-16$ (Fig. 3B). The expression distribution among different smORF classifications was also compared, as shown in Fig. 3C.

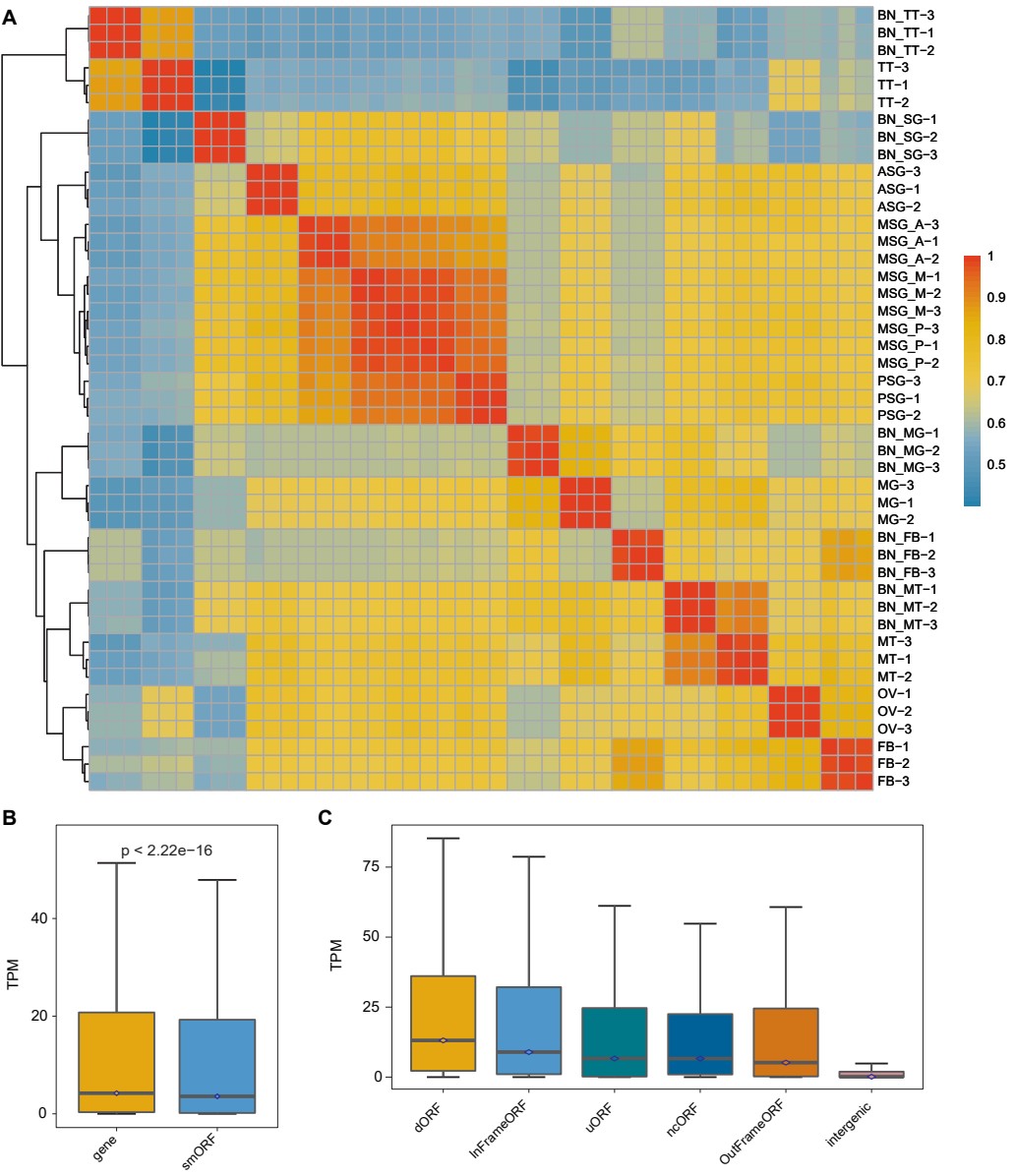

**Figure 3   The expression pattern of smORF in the silkworm.** (A) Heatmap of sample correlation matrix based on smORF expression. (B) Comparison of expression levels between protein-coding gene and smORF. (C) Comparison of expression distribution among different smORF classifications.

Based on the expression results, we analyzed the differentially expressed smORFs between different silkworm tissues with FDR $\leq$ 0.05 and log2(fold change) $\geq$ 2 as the screening condition. As shown in Table 2, the average number of smORFs differentially expressed among organizations was 5874.5, of which the largest number was between silk gland and testis. The number of DES (differential-expressed smORFs) reached 12,454, of which the smallest number was between the middle part of the middle silk gland and the posterior

**Table 2  The number of differentially expressed smORFs between tissues.**

|  | ASG | FB | MG | MSG_A | MSG_M | MSG_P | MT | OV | PSG | SG | TT | Average |
|---|---|---|---|---|---|---|---|---|---|---|---|---|
| ASG | 0 | 4,686 | 5,159 | 3,260 | 3,589 | 3,514 | 4,379 | 5,156 | 4,148 | 9,582 | 8,545 | 5201.8 |
| FB | 4,686 | 0 | 4,103 | 4,356 | 4,878 | 4,772 | 4,133 | 5,201 | 5,098 | 8,685 | 8,186 | 5409.8 |
| MG | 5,159 | 4,103 | 0 | 4,925 | 5,413 | 5,297 | 3,339 | 6,785 | 5,649 | 9,053 | 8,839 | 5856.2 |
| MSG_A | 3,260 | 4,356 | 4,925 | 0 | 961 | 1,149 | 4,465 | 5,120 | 1,712 | 7,919 | 8,324 | 4219.1 |
| MSG_M | 3,589 | 4,878 | 5,413 | 961 | 0 | 171 | 5,073 | 5,794 | 1,155 | 8,461 | 8,965 | 4446 |
| MSG_P | 3,514 | 4,772 | 5,297 | 1,149 | 171 | 0 | 4,935 | 5,761 | 907 | 8,677 | 9,010 | 4419.3 |
| MT | 4,379 | 4,133 | 3,339 | 4,465 | 5,073 | 4,935 | 0 | 5,986 | 5,183 | 9,112 | 8,579 | 5518.4 |
| OV | 5,156 | 5,201 | 6,785 | 5,120 | 5,794 | 5,761 | 5,986 | 0 | 6,365 | 12,340 | 6,219 | 6472.7 |
| PSG | 4,148 | 5,098 | 5,649 | 1,712 | 1,155 | 907 | 5,183 | 6,365 | 0 | 8,570 | 9,003 | 4779 |
| SG | 9,582 | 8,685 | 9,053 | 7,919 | 8,461 | 8,677 | 9,112 | 12,340 | 8,570 | 0 | 12,454 | 9485.3 |
| TT | 8,545 | 8,186 | 8,839 | 8,324 | 8,965 | 9,010 | 8,579 | 6,219 | 9,003 | 12,454 | 0 | 8812.4 |

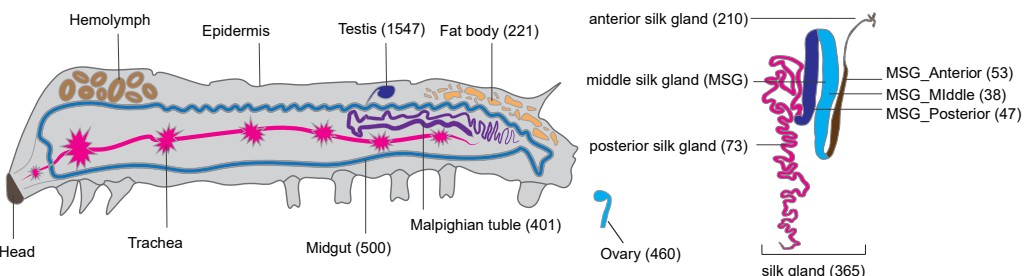

**Figure 4  Tissue-specific expressed smORFs genes in the silkworm (TSI threshold = 1.0).**

part of the middle silk gland (only 171). In general, the proximity of the differentiation direction of tissue samples was related to the reduction in the difference smORFs.

## Tissue-specific expression of smORFs

Like tissue-specific genes (called luxury genes), tissue-specific smORFs may also play a key role in specific physiological functions of their respective tissues (*Nguinkal et al., 2021*; *He et al., 2018*). To explore the tissue-specific expression of smORFs in different tissues, TSPEX software was used to analyze the tissue specificity of all identified smORFs, and a TSI value was used to indelicate and filter the tissue-specificity. When the TSI threshold was set to 0.8, the largest number of smORFs specifically expressed in each tissue was in testis, which was 1,702, and the smallest number was in the middle part of the middle silk gland, which was 49, with an average number of 444. When the TSI threshold was set to the maximum value of 1.0, the number of tissue-specific expression smORFs decreased in general, but the maximum value was still the testis. The total number decreased to 1,547 and the smallest number remained in the middle part of the middle silk gland, which decreased to 38, and the average number also decreased to 356 (Fig. 4, Table S8).

## DISCUSSION
## Comprehensive smORF candidates

The data analyzed in this paper were downloaded from 45 samples of silkworm strains p50T and o751, including 10 tissues, and the same or similar tissues also showed a strong correlation between strains. These data covered most of the major silkworm tissues and silkworm genes, providing a prerequisite for the accuracy and completeness of subsequent definition and analysis of smORFs. In the smORFs identification steps, we first constructed the silkworm-specific Kozak sequences, and then used ORFfinder to predict the assembled transcript sequences and the conserved sequences between related species. By integrating these previous reliable methods, a pipeline for smORFs identification and analysis was established.

Systematic studies and analyses of smORFs have been conducted in many eukaryotes, such as *Arabidopsis,* yeasts, *Drosophila* and mouse (*Frith et al., 2006*). The research methods and results from these species, especially fruit flies that belong to the Class Hexapoda, have great reference values for the study of silkworm. According to our results, *B. mori*, like *Drosophila* and other species, have a large number of different types of smORFs. The total number of predicted smORFs was larger than reported in previous studies, possibly because we integrated a variety of evidences. Our analysis showed that there were at least 738 functional smORFs in *B. mori*, supported by the array of evidencs presented here, accounting for 4.59% (738/16,069) of the 16,069 coding genes (*Lu et al., 2020*). The ratio of the number of functional smORFs to the number of coding genes was consistent with previous studies. *Hanada et al. (2007)* believe that there may be 3,241 smORFs in *Arabidopsis*, this is about 5% of the *Arabidopsis* genes. According to the study of *Basrai, Hieter & Boeke (1997)* and *Frith et al. (2006)*, the functional smORFs in yeast and mice also account for about 5% of their genes. In a later study of fruit flies, *Ladoukakis et al. (2011)* pointed out that there were at least 401 (3% of total genes) functional smORFs in *Drosophila* and speculated that the number should be no less than 4,561.

## Role of credibility classification

Following the protocols from *Couso & Patraquim (2017)* and *Wu et al. (2019)*, we divided the identified 34,401 smORFs into six types according to their position in the genome and their relative relationship with genes, namely InFrameORF, OutFrameORF, dORF, intergenicORF, ncORF, and uORF. We further classified them into high confidence and low confidence levels according to the supporting evidence obtained for each smORF. The low confidence smORFs identified in this paper were supported by either silkworm-specific Kozak sequences and transcriptome evidence or silkworm-specific Kozak sequences and conserved genome sequence of related species; while the smORFs with high confidence levels should have at least one of the supporting lines of evidence indicating that they were in or with homologous sequences in the database, or must be able to be identified by CPPred-sORF as having coding potential. Without doubt, we may miss some real smORFs (false negatives), and there are some false positives in our smORFs candidates. Our classification system based on credibility would facilitate downstream experimental utilization, especially when we need to study one or several smORFs, utilizing the most robust estimation type (HC1).

### The feasibility of smORF function researches

Differential expression analyses and tissue-specific expression analyses are often used to narrow the range of candidate genes to provide conditions for studying the function and mechanism of genes. For differentially expressed genes, the absolute expression and relative expression levels of genes were both important screening conditions. For tissue-specific genes, except for their expression levels, tissue specificity was also an important reference. For smORFs with tissue-specific expression or inter-tissue differential expression, their functions are more likely to be important and related to their corresponding tissues.

## CONCLUSIONS

SmORFs, as one of the important components of the genome, play a critical regulatory role in a series of processes. Using transcriptomics and genomics data, we found at least 738 highly reliable smORFs in *B. mori*, and an additional 34,401 smORFs that were partially supported. These numbers are similar to those found in other organisms. However, we should note that, with more available datasets from different technologies, such as translation-omics and proteomics, this list of smORF candidates may be extend, which urges us generating more publicly available genetic resources for this species with great biomedical and industrial importance. Altogether, the researches on smORFs in *B. mori* may help deepen our understanding of smORFs, so as to provide the guideline for subsequent studies of smORFs in other species.

## ACKNOWLEDGEMENTS

We are grateful to Prof. Jinshu Xiao from Sichuan Academy of Agricultural Sciences for helpful discussions on topics related to this work. We would like to thank previous researchers and communities to submit the genome and transcriptome data of the domestic silkworm to NCBI SRA databases.

### Funding

This work was supported by the China Agriculture Research System project (NO. CARS-18-SYZ19), the Sichuan Province '14th Five-Year Plan' crop and livestock breeding projects (NO. 2021YFYZ0024) and the Sichuan Province Finance Independent Innovation project (NO. 2022ZZcx084). The funders had no role in study design, data collection and analysis, decision to publish, or preparation of the manuscript.

### Grant Disclosures

The following grant information was disclosed by the authors:
China Agriculture Research System: CARS-18-SYZ19.
Sichuan Province '14th Five-Year Plan: 2021YFYZ0024.
Sichuan Province Finance Independent Innovation: 2022ZZcx084.

## Competing Interests
Ziyan Huang & Cao Deng are employed by LyuKang; DNA Stories Bioinformatics Center

## Author Contributions

- Linrong Wan conceived and designed the experiments, analyzed the data, prepared figures and/or tables, authored or reviewed drafts of the article, and approved the final draft.
- Wenfu Xiao conceived and designed the experiments, performed the experiments, prepared figures and/or tables, authored or reviewed drafts of the article, and approved the final draft.
- Ziyan Huang analyzed the data, prepared figures and/or tables, and approved the final draft.
- Anlian Zhou performed the experiments, prepared figures and/or tables, and approved the final draft.
- Yaming Jiang performed the experiments, authored or reviewed drafts of the article, and approved the final draft.
- Bangxing Zou performed the experiments, prepared figures and/or tables, and approved the final draft.
- Binbin Liu performed the experiments, prepared figures and/or tables, and approved the final draft.
- Cao Deng conceived and designed the experiments, analyzed the data, authored or reviewed drafts of the article, and approved the final draft.
- Youhong Zhang conceived and designed the experiments, authored or reviewed drafts of the article, and approved the final draft.

## DNA Deposition
The following information was supplied regarding the deposition of DNA sequences:

The smORFs candidates and differentially expressed and tissue-specific smORFs are available at Github: https://github.com/dengcao3/silkworm_smORF.

## Data Availability
The smORFs candidates and differentially expressed and tissue-specific smORFs are available at Github: https://github.com/dengcao3/silkworm_smORF.

## Supplemental Information
Supplemental information for this article can be found online at http://dx.doi.org/10.7717/peerj.14682#supplemental-information.

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
