# Peer review of "Systematic identification of smORFs in domestic silkworm (Bombyx mori)"

_PeerJ, doi:10.7717/peerj.14682_

## Round 0.1 · original submission · Major Revisions

Your manuscript has been reviewed by two experts in the field. As you can see from their comments below, both of them basically admit the value of this work but raise several points for further improvement. Amongst them, reviewer 1 asks to add a discussion on the comparison with known Drosophila examples, and reviewer 2 points out the possibility of detecting many false positives. I think that answering the point by reviewer 1 would be useful to answer the point by reviewer 2. Perhaps, the authors should also note the inherent limitation of their approach. Please read their comments carefully and revise the manuscript accordingly.

Reviewer 1 ·

Basic reporting

The manuscript entitled “Systematic identification of smORFs in domestic silkworm (Bombyx mori)” is straightforward and well-written. I recommend that the authors compare the identified Bombyx small open reading frames (smORFs) to known Drosophila smORFs and add some discussion. For example, can some of the functional smORFs in Drosophila be found in B. mori? Differences in the number of smORFs between organisms should be also discussed in the Discussion section. Features of the Bombyx smORFs are useful to researchers, but their information is limited. The authors should provide or share the raw data (i.e., assembled sequences and the expression data) for further research in this field. Providing the raw data would also increase the credibility of this study (also see PeerJ policy).

Experimental design

The background and purpose of this study are clear. The strategy for smORF identification is well considered and described in detail in the Materials and Methods section.

Validity of the findings

Using public genomic and transcriptomic data, Wan et al. systematically and comprehensively identified smORFs in the domesticated silkworm, B. mori, which is an economically important lepidopteran insect. Because there is little information on smORFs of B. mori and also other lepidopteran insects, this study is very informative for insect scientists.

Additional comments

L50: Move references 13 and 14 after "silk production" on line 49. Add appropriate references here.
Introduction: References 18 and 19 are missing.
L79: B. mori strain o751 was downloaded from the NCBI Genomic and transcriptomic data resources → Genomic and transcriptomic data of B. mori were downloaded from NCBI.
L86-87: Why the authors use these lepidopteran insects (Melitaea cinxia and Operophtera brumata) for the conserved region prediction?
L91: B. mori strain P50T → B. mori (italic) strain p(lower case)50T
L94: The RNA-seq data of B. mori → The RNA-seq data of B. mori strain o751 (wild-type)
L307: B. mori → italic
L308-309: Bombyx mori → B. mori (italic)
L364: Reference 19 is missing in the text. This reference was published in Nature Communications.
Figure 1A: conservative region → conserved region?
Figure 1D: For example, 10 → +1, 9 → -1 are good.

Reviewer 2 ·

Basic reporting

no comment

Experimental design

In this manuscript, the authors try to comprehensively identify small open reading frames (smORFs) encoded in silkworm (Bombyx mori) genome by means of various bioinformatics methods.

Identification of real smORFs is still a challenging theme. Real smORFs must be translated, or at least bound by translational machinery. Presently two experimental procedures, i.e., ribosome profiling and mass-spectrometry-based proteomics analysis, are known to be most effective approaches for this purpose. The study presented in the manuscript relies on only bioinformatics analyses without taking account of the benefits of these useful procedures.

Validity of the findings

Therefore, it is difficult to assess how much fraction of identified smORFs is real and how much is false positive. Apart from this fundamental limitation, the bioinformatics analyses seem to be mostly done adequately. The manuscript is generally well organized to preset the authors’ findings.

Additional comments

However, I have several concerns about some parts of the manuscript.
1) Lines 129-131. The sentence is ambiguous. I don’t understand what ‘replace’ means.
2) Lines 137-138. The latter half of this sentence is also not understandable for me.
3) Overall, how does the silkworm-specific Kazak sequence look like? How does it resemble or differ from those of other species and the smORF-derived Sequence Logo shown in Figure 1E?
4) Intuitively, smORF-derived Kozak sequence (Fig. 1E) appears to be too ‘strong’ compared with species-specific Kozak sequences so far published. In particular, complete conservation at -4, -3 and +4 sites is rather surprising. It should be noticed that this observation is not an artifact due to the smORF-finding procedure.
5) Author names of Reference No 5, 7, 15, 21, 33, 38-41, 45, 47, and 48 are incomplete.

---

## Round 0.2 · Minor Revisions

Your revised manuscript has been reviewed by the two original reviewers. Though both of them agree to accept your manuscript, they point out a number of minor points. Please read their comments carefully and improve your manuscript.

Reviewer 1 ·

Basic reporting

In the revised manuscript, the authors have improved the manuscript following to the suggestions of all the reviewers. Therefore, I think this revised manuscript would be accepted in PeerJ after some minor corrections.

Experimental design

no comment

Validity of the findings

no comment

Additional comments

L15: Bombyx mori → Bombyx mori (italic)
L45: ,metabolism → ,(space)metabolism
L66: identifying smORFs → identifying open reading frames (ORFs)? If so, please also change “open reading frames (ORFs)” in L67-68 to “ORFs”.
L79: Genomic and transcriptomic data of B. mori were downloaded from NCBI. → Genomic and transcriptomic data
L98-99: Trimming and alignment of RNA-Seq reads to silkworm reference genome and reference genome-guided transcript assembly → “Sequencing alignment and transcript assembly” might be better.
L129: ,the first codon → ,(space)the first codon
L142: Drosophila melanogaster → Drosophila melanogaster genome?
L148 are consistent with → were consistent with
L317: arabidopsis, yeasts, drosophila → Arabidopsis, yeasts, Drosophila
L355-357: Why? Please cite some references.
Legend for Figure 1: (C) Composition of smORFs with a high confidence level. → Please explain “CPPred-sORF” and “Homology”.
Table S1: The cited references [2-5] are incorrect (Yokoi et al. (2021) also cite incorrect references). See references below.
Ichino, F.; Bono, H.; Nakazato, T.; Toyoda, A.; Fujiyama, A.; Iwabuchi, K.; Sato, R.; Tabunoki, H. Construction of a simple evaluation system for the intestinal absorption of an orally administered medicine using Bombyx mori larvae. Drug Discov. Ther. 2018, 12, 7–15.
Kobayashi, Y.; Nojima, Y.; Sakamoto, T.; Iwabuchi, K.; Nakazato, T.; Bono, H.; Toyoda, A.; Fujiyama, A.; Kanost, M.; Tabunoki, H. Comparative analysis of seven types of superoxide dismutases for their ability to respond to oxidative stress in Bombyx mori. Sci. Rep. 2019, 9, 2170.
Kikuchi, A.; Nakazato, T.; Ito, K.; Nojima, Y.; Yokoyama, T.; Iwabuchi, K.; Bono, H.; Toyoda, A.; Fujiyama, A.; Sato, R. Identification of functional enolase genes of the silkworm Bombyx mori from public databases with a combination of dry and wet bench processes. BMC Genom. 2017, 18, 1–12.

Reviewer 2 ·

Basic reporting

no comment

Experimental design

no comment

Validity of the findings

no comment

Additional comments

I think the manuscript has been significantly improved. In particular publicize the authors findings is highly appreciated.

I have only minor comments.
1) Line 325. Kouske -> Hamada.
2) Line 333. Fllowing -> Follwing.
3) Line 365. Correct the sentence.
4) Figure legend 1 (D). If the logo was derived from ordinary genes of B. mori as the authors claim in the response letter, this legend is misleading and should be modified accordingly. Preferably, sequence logos of Kozak motifs of smORFs of different confidence levels (say HC1-3) should be presented along with that shown here.

---

## Round 0.3 · accepted · Accept

Since I confirmed that the authors have addressed all of the minor points addressed by reviewers, I am happy to recommend its acceptance to the section editor. Congratulations!